# What we can and cannot see from the surveillance for drug-resistant *Pseudomonas aeruginosa*—Findings from the evaluation of a surveillance system for multidrug-resistant *P. aeruginosa* infections in Japan

Shogo Otake[1,2], Takuya Yamagishi[3,4]*, Takayuki Shiomoto[2], Manami Nakashita[3,4], Hitomi Kurosu[3,4], Chiaki Ikenoue[4], Hirofumi Kato[4], Munehisa Fukusumi[4], Tomoe Shimada[4], Takuri Takahashi[5], Motoi Suzuki[5], Teruo Kirikae[6], Yoshichika Arakawa[7], Kandai Nozu[1], Tomimasa Sunagawa[4], Motoyuki Sugai[3]

1 Department of Pediatrics, Kobe University Graduate School of Medicine, Hyogo, Japan, 2 Field Epidemiology Training Program, National Institute of Infectious Diseases, Tokyo, Japan, 3 Antimicrobial Resistance Research Center, National Institute of Infectious Diseases, Tokyo, Japan, 4 Center for Field Epidemiology Intelligence, Research, and Professional Development, National Institute of Infectious Diseases, Tokyo, Japan, 5 Center for Surveillance, Immunization, and Epidemiologic Research, National Institute of Infectious Diseases, Tokyo, Japan, 6 AMR Research Laboratory, Juntendo Advanced Research Institute for Health Science, Juntendo University, Tokyo, Japan, 7 Department of Microbiology, Fujita Health University School of Medicine, Aichi, Japan

* tack-8@niid.go.jp

## Abstract

### Introduction

Antimicrobial resistance in *Pseudomonas aeruginosa* is one of the global health concerns. Like many countries, Japan monitors multidrug-resistant *P. aeruginosa* (MDRP) infections through a national sentinel surveillance system, which has shown a recent decline in MDRP reports. We evaluated this surveillance system to verify the validity of this trend and explore future directions.

### Methods

We conducted a cross-sectional mixed-method study based mainly on the guidelines published by the United States Centers for Disease Control and Prevention in 2001. As a quantitative method, we analyzed characteristics of reports on MDRP infections from designated sentinel sites (DSSs) between 2013–2022. A questionnaire was sent to identifiable DSSs (target DSSs) requesting data on accurate numbers of MDRP infections between 2018–2022 to assess attributes such as geographical representativeness. Additionally, we conducted as a qualitative method face-to-face, semi-structured key informant interviews with surveillance system stakeholders to assess its usefulness and challenges.

**Data availability statement:** All relevant data are within the paper and its Supporting Information files.

**Funding:** This research was supported by the Ministry of Health, Labour and Welfare, Japan [Grant number 23HA1004]. The funder had no role in study design, data collection and analysis, decision to publish, or preparation of the manuscript.

**Competing interests:** The authors have declared that no competing interests exist.

**Abbreviations:** AMR, Antimicrobial resistance; AR Lab Network, Antibiotic Resistance Laboratory Network; CDC, Centers for Disease Control and Prevention; CLSI, Clinical and Laboratory Standards Institute; CRPA, Carbapenem-resistant *Pseudomonas aeruginosa*; DSS, Designated sentinel site; DTRP, Difficult-to-treat resistant *Pseudomonas aeruginosa*; EARS-Net, European Antimicrobial Resistance Surveillance Network; EUCAST, European Committee on Antimicrobial Susceptibility Testing; JANIS, Japan Nosocomial Infections Surveillance; MDRP, Multidrug-resistant *Pseudomonas aeruginosa*; MRSA, Methicillin-resistant *Staphylococcus aureus*; NAP, National action plan; NESID, National Epidemiological Surveillance of Infectious Diseases; PDRP, Pan drug-resistant *Pseudomonas aeruginosa*; PPV, Positive predictive value; TDSS, Target designated sentinel site; U.S., United States; WHO, World Health Organization; XDRP, Extensively drug-resistant *Pseudomonas aeruginosa*

## Results

From 2013 to 2022, 1,666 cases of MDRP infections were reported by 463 target DSSs, which were scattered across the county. We obtained valid responses to the questionnaire survey from 231 target DSSs (49.9%). From 2018 to 2022, these sites reported 277 cases as MDRP infections, while 184 cases were accurate cases of MDRP infection, with both numbers declining over time. False reporting and under-reporting of MDRP infections were common, resulting in a positive predictive value of 0.45 and a sensitivity of 0.65 for the reports of MDRP infections to the surveillance system. The interviews highlighted the difficulties in timely detection, accurate reporting, and international data comparison.

## Conclusion

Our evaluation indicated that the current sentinel surveillance system for MDRP infections partially captured the true decreasing trend in Japan. However, as the epidemiology of drug-resistant *P. aeruginosa* is changing, national policy and surveillance strategies would need to address changing public health needs.

## Introduction

Bacterial antimicrobial resistance (AMR) is a global concern and a significant public health threat. A comprehensive assessment of the global burden of AMR based on data from 204 countries and territories estimated 4.71 million deaths in 2021. Additionally, it is projected that by 2050, AMR could be associated with 8.22 million deaths globally, including 1.91 million directly attributable to AMR [1]. In Japan, although no studies have estimated the total number of AMR-related infections or deaths nationwide, the disease burden has been reported to be significantly higher in AMR infections including community-acquired infections compared to non-AMR infections [2,3]. After the release of the national action plan (NAP) on AMR, use of antimicrobials decreased between 2015 and 2021. Nevertheless, the disease burden of bloodstream infections caused by major AMR pathogens has not changed significantly [4].

*Pseudomonas aeruginosa* is a Gram-negative bacterium responsible for opportunistic infections in immunocompromised hosts [5] that is frequently associated with healthcare-acquired infections [6]. Additionally, it is also a major public health concern, as evidenced by the outbreak linked to artificial tears in the United States (U.S.) in 2023 [7].

AMR of *P. aeruginosa* is recognized as a global health concern. *P. aeruginosa* has multiple mechanisms for acquiring antibiotic resistance [8,9] that can lead to a serious prognosis [5,6]. In particular, carbapenem-resistant *P. aeruginosa* (CRPA) and multidrug-resistant *P. aeruginosa* (MDRP) are associated with higher mortality rates compared to susceptible strains of *P. aeruginosa* [5,10,11]. Thus, the World Health Organization (WHO) listed CRPA as a pathogen of critical priority in 2017. Although CRPA has been moved from critical to high priority in the 2024 update,

the need for innovative research and development of antibiotics remains due to the significant burden in high-income countries and several world regions [12,13]. The U.S. Centers for Disease Control and Prevention (CDC) has estimated 32,600 cases and 2,700 deaths due to MDRP infections annually and has classified them as a serious public health threat, calling for the need for prompt and sustained action in 2019 [14]. Furthermore, data published in 2024 showed that MDRP infections increased in 2021–2022 compared to 2019, despite a previously observed decreasing trend [15].

Some countries and regions conduct surveillance for drug-resistant *P. aeruginosa*. The European Antimicrobial Resistance Surveillance Network (EARS-Net) collects AMR data from European Union and European Economic Area countries through the European Surveillance System and annually reports the proportions of MDRP within the WHO European Region and CRPA in each country [16]. In the U. S., the CDC tracks carbapenemase enzymes in CRPA by using data generated by the Antibiotic Resistance Laboratory Network (AR Lab Network) and CDC laboratories [17,18].

Japan has two national-level surveillance system for drug-resistant *P. aeruginosa* [19]. First, the National Epidemiological Surveillance of Infectious Diseases (NESID) was launched in April 1999 based on the "Act on the Prevention of Infectious Diseases and Medical Care for Patients with Infectious Diseases". In this system, the sentinel sites report the MDRP infections with the information of age, sex, and detected specimen to the local public health center monthly to assess the nationwide trend in the number of MDRP infections. The central infectious disease surveillance center (National Institute of Infectious Diseases) integrates the data and shares it with the Ministry of Health, Labour and Welfare (S1 Fig) [20,21]. Second, the Japan Nosocomial Infections Surveillance (JANIS) was implemented in 2000 as a voluntary surveillance system designed to specifically monitor infection and colonization in healthcare settings. This system is a national statistical survey based on the "Statistics Act" and has collected all routine microbiological test results for more than nine million clinical samples annually from over 2,400 facilities as of 2023 in a data format that is uniform even though each hospital uses a different AMR testing system [19,22–24]. In both NESID and JANIS, MDRP is defined as *P. aeruginosa* resistant to three anti-pseudomonas antibiotics: carbapenems, amikacin, and fluoroquinolones [21]. The susceptibility criteria used by NESID and JANIS for sentinel surveillances differ somewhat from the clinical breakpoints established by the Clinical and Laboratory Standards Institute (CLSI) and the European Committee on Antimicrobial Susceptibility Testing (EUCAST) (S1 Table) [25,26]. These discrepancies may often cause confusion among clinicians and lead to inaccurate reporting.

The number of MDRP infections reported from sentinel sites to NESID peaked in 2003 at 759 cases (1.62 per sentinel site). Since then, the number has decreased, with 103 cases reported in 2022 (0.22 per sentinel site) [27]. It is thus necessary to evaluate whether this trend accurately represents the actual trend in MDRP infections nationwide. In 2023, the AMR NAP in Japan set a goal of achieving carbapenem resistance of *P. aeruginosa* of 3% or less by 2027 rather than focusing on the occurrence of MDRP [28].

Our study aimed to evaluate whether NESID accurately captures the nationwide trend in MDRP infections and determine possible challenges related to inaccuracy, if any, for better control of drug-resistant *P. aeruginosa* through improvement of the current surveillance system.

## Methods

### Study design

We conducted a cross-sectional study in 2023 using a mixed-methods approach that incorporated both quantitative and qualitative analyses. We assessed the nine attributes of representativeness, positive predictive value (PPV), sensitivity, data quality, acceptability, timeliness, simplicity, flexibility, and stability, and discussed the usefulness of this surveillance (Table 1). This evaluation was based mainly on the Guidelines for Evaluating Public Health Surveillance Systems published by the U.S. CDC in 2001 [29]. Additionally, on the basis of this evaluation, we listed the advantages and disadvantages of possible surveillance systems of drug-resistant *P. aeruginosa* infections.

**Table 1. Quantitative and qualitative evaluation for attributes of the surveillance system for MDRP infections in Japan.**

| Attributes | Perspectives | Rationales | Results |
|---|---|---|---|
| Quantitative evaluation | | | |
| Representativeness | Are TDSSs selected without bias based on region? | The percentage of TDSSs among all medical institutions and the percentage of TDSSs with over 300 or more beds among all TDSSs were similar across prefectures (Fig 1). | Yes |
| PPV and Sensitivity | Do the trend in the number of reports to NESID as MDRP infections represent the trend in the number of accurate MDRP infections at TDSSs? | From 2018 to 2022, both the number of reported cases as MDRP infections and the accurate number of MDRP infections decreased over time (Fig 2). | Yes |
| PPV | Is the percentage of accurate MDRP infections among cases reported to NESID as MDRP infections at TDSSs high enough to be utilized? | PPV at subjected sentinel sites was 0.45, with 0.19 at sites with fewer than 300 beds and 0.48 at sites with 300 or more beds (Table 3). | No |
| Sensitivity | Is the percentage of reported cases to NESID among accurate MDRP infections at TDSSs high enough to be utilized? | Sensitivity at TDSSs was 0.65, with 1.00 at sites with fewer than 300 beds and 0.64 at sites with 300 or more beds (Table 3). | Yes and No |
| Data quality | Is the data recorded in the surveillance system complete and valid? | For 1,670 cases reported to NESID as MDRP infections from 2013 to 2022, all required items (age, sex, and detected specimen) were registered. | Yes |
| Qualitative evaluation | | | |
| Representativeness | (i) Does each prefecture select DSSs appropriately based on criteria? | In the subjected prefecture, nine DSSs were distributed evenly across the region. All DSSs offer both internal medicine and surgery, and eight had 300 and more beds. | Yes |
| | (ii) Are DSSs sufficient for reporting MDRP infections? | Reporting solely from DSSs would be inadequate for monitoring the outbreaks of MDRP infections because outbreaks occurring at long-term care facilities and small medical facilities would be overlooked. | No |
| Acceptability | Do the DSSs accept reporting of cases monthly? | It was easy to report MDRP infections once a month. However, physicians questioned whether monthly reports of MDRP infections, which require an outbreak response, were sufficient. | No |
| Timeliness | Is information promptly provided to the public after a physician diagnoses a patient? | As MDRP infections are reported once a month, prompt reports after diagnosis and feedback to the public are not required. | Not applicable |
| Simplicity | (i) Is it easy to report and confirm a case by using the system? | It was easy for physicians to report cases to NESID and information flow is simple, but staff at local infectious disease information centers found it difficult to confirm the accuracy of these reports. | Yes and No |
| | (ii) Is the case definition easy to understand? | The case definition is complicated due to the need to confirm resistance to three drugs using standards different from the breakpoints of CLSI and EUCAST, as well as the need to exclude colonized cases. | No |
| PPV | Does the system effectively prevent incorrect or missing reports? | There was a possibility of reporting errors caused by complicated case definition. | No |
| Flexibility | (i) Can the case definition be changed easily? | The case definition is strictly governed by Japanese law, requiring the approval of numerous experts and authorities based on clear evidence to be changed. | No |
| | (ii) Can DSSs add or change the reporting contents freely? | The reporting contents are also strictly governed by Japanese law. | No |
| | (iii) Can the prefecture change the DSSs easily? | The governor has the authority to change DSSs, but these sites have remained unchanged for at least 10 years in the subjected prefecture. | Yes and No |
| Stability | Can the surveillance system continue to function after unexpected accidents? | It is rare for the surveillance system's computers to experience unscheduled outages and downtime. In addition, this system continues to function unless directly affected by large-scale disasters such as the Japan Earthquake and Tsunami in 2011 and the Noto Earthquake in 2024. | Yes |

*(Continued)*

**Table 1.** (Continued)

| Attributes | Perspectives | Rationales | Results |
|---|---|---|---|
| Usefulness | Do government and medical institutions utilize the system to combat AMR? | The system makes it difficult for us to detect and respond to a cluster of MDRP infections at facilities other than DSSs. Additionally, the National Action Plan in Japan is aimed at reducing CRPA infections, not MDRP infections, and comparison of the situation with other developed countries would be difficult due to different definitions of MDRP. | No |

AMR, antimicrobial resistance; CLSI, Clinical and Laboratory Standards Institute; DSS, designated sentinel site; CRPA, carbapenem-resistant *Pseudomonas aeruginosa*; EUCAST, European Committee on Antimicrobial Susceptibility Testing; MDRP, multidrug-resistant *Pseudomonas aeruginosa*; NESID, National Epidemiological Surveillance of Infectious Diseases; PPV, positive predictive value; TDSS, target designated sentinel site.

This evaluation was based on the Guidelines for Evaluating Public Health Surveillance Systems published by the Centers for Disease Control and Prevention in 2001 [29].

## NESID for MDRP infections in Japan

In Japan, which is divided into 47 prefectures, each prefectural government selects at least one designated sentinel site (DSS) per secondary medical area, which is a medical administrative area, regardless of the type of healthcare facility, such as tertiary hospitals or secondary health centers, under the Medical Care Law. The DSS generally is capable of hospitalizing at least 300 patients and has internal medicine and surgery departments, from which particular AMR bacterial infections including MDRP infections are reported [20]. Reporting of AMR infections from DSSs is done by the hospital administrators. The DSSs can be changed according to the local governments' needs. As details of the DSSs are not publicly available, we defined them as target DSSs (TDSSs) that remained open as of December 2022 and met either of the following two conditions:

(a) DSSs that reported MDRP infections to NESID from January 2013 to December 2022, or

(b) DSSs that reported methicillin-resistant *Staphylococcus aureus* (MRSA) infections to NESID from January to December 2021.

Condition (b) was required because MRSA infections are reported from most of the hospitals in Japan [23,24].

## Data collection

**Quantitative evaluation.** We analyzed the characteristics of reports as MDRP infections to NESID from DSSs between January 2013 and December 2022. We accessed the NESID data for research purposes on January 11, 2023. On February 1, 2023, we sent a questionnaire that we had developed (S1 Appendix) to the administrators or infection control teams in the TDSSs to request the accurate number of MDRP infections occurring at each site between January 2018 and December 2022. The questionnaire was paper based and written in Japanese. If responses from a TDSS included the answer "unknown" for any year, we excluded all responses from that TDSS as invalid responses. We evaluated these results separately for hospitals with 300 or more beds and hospitals with fewer than 300 beds.

**Qualitative evaluation.** On the basis of the information flow (S1 Fig), we conducted face-to-face, semi-structured key informant interviews with 17 stakeholders of the surveillance system between December 2, 2022, and August 24, 2023, to ask about its relevant attributes such as representativeness, acceptability, simplicity, and usefulness. In addition to three representatives from national academic authorities, our interviewees included two public health officers working at one local infectious disease surveillance center in a specific prefecture. We also selected six physicians including infectious diseases specialists, two infection control nurses, two pharmacists, and two clinical microbiology technologists working at five hospitals within the same prefecture (two TDSSs and three non-TDSSs), which were introduced by each local government. Well-trained interviewers conducted each interview by using interview guides (S2 Appendix) that were

reviewed by experienced physicians and clinical microbiology technologists. Although we did not electronically record the interviews, at least two well-trained interviewers conducted each interview to ensure that no details were missed.

## Analysis

We calculated the mean age with standard deviation and sex distribution of the patients using the NESID data and calculated the distribution of TDSSs as a percentage of all hospitals in each prefecture based on public data [30] and presented the distribution of TDSSs that are hospitals with 300 beds or more. Regarding the questionnaire survey, we defined (i) as the number of reports of MDRP infections to NESID from TDSSs between January 2018 and December 2022 and (ii) as the number of MDRP infections reported as accurate MDRP infections on the survey. We defined (iii) as the number of cases misreported as MDRP infections despite not meeting the criteria and (iv) as the number of underreported accurate MDRP infections computed as follows (S2 Fig):

(iii) = (i) − (ii) ((i)> (ii))

(iv) = (ii) − (i) ((ii)> (i))

We counted (i) through (iv) for each hospital, aggregated them annually, and evaluated the trend. In addition, we calculated the PPV and sensitivity of reports of MDRP infections by using pooled data in NESID from 2018 to 2022. R version 4.4.3 was used for all of the calculations [31].

## Ethics considerations

Because the information handled in this study cannot lead to the identification of individuals, the National Institute of Infectious Diseases did not require informed consent or ethical review (receipt no. 1478).

## Results

In total, 1,670 cases were reported to NESID as MDRP infections from 347 DSSs between 2013 and 2022 (S2 Table). The mean age (± standard deviation) of the reported patients was 71.7±17.6 years, and 1,168 patients (69.9%) were male. The number of cases reported as MDRP infections showed a decreasing trend for both sexes, especially among the elderly (S3 Fig and S2 Table). One DSS reporting two cases closed in 2015, and the status of two others reporting one case each remained unknown. Although 119 DSSs did not report any cases as MDRP infections during the same period, they reported at least one case of MRSA infection in 2022. Thus, TDSSs in this study were defined as 463 facilities reporting 1,666 cases.

All 463 of the TDSSs offer both internal medicine and surgery. Among them, 347 (74.9%) have 300 or more beds and 116 (25.1%) have fewer than 300 beds, accounting for 24.2% and 1.7%, respectively, of all medical institutions with the same capacities in Japan [32]. From 2013 to 2022, at least one case was reported as a MDRP infection by 262 (75.5%) of the 347 TDSSs with 300 or more beds and 82 (70.7%) of 116 TDSSs with fewer than 300 beds. The results of the quantitative and qualitative evaluation for each attribute are summarized in Table 1.

## Quantitative evaluation

TDSSs accounted for 5–10% of all medical institutions in many prefectures, and TDSSs with 300 or more beds accounted for more than 80% of all TDSSs in most prefectures (Fig 1 and S3 Table). For the questionnaire survey, we obtained valid responses from 231 TDSSs (49.9%) (Tables 2 and S4). The respondents were primarily clinical microbiology technologists (55.9%), followed by nurses (28.0%), doctors (5.9%), and pharmacists (2.1%). From 2018 to 2022, these TDSSs reported 277 cases as MDRP infections, while 184 cases were accurately reported MDRP infections, with both numbers decreasing over time (Fig 2 and S5 Table). There were 162 cases misreported as MDRP infections despite not meeting the criteria

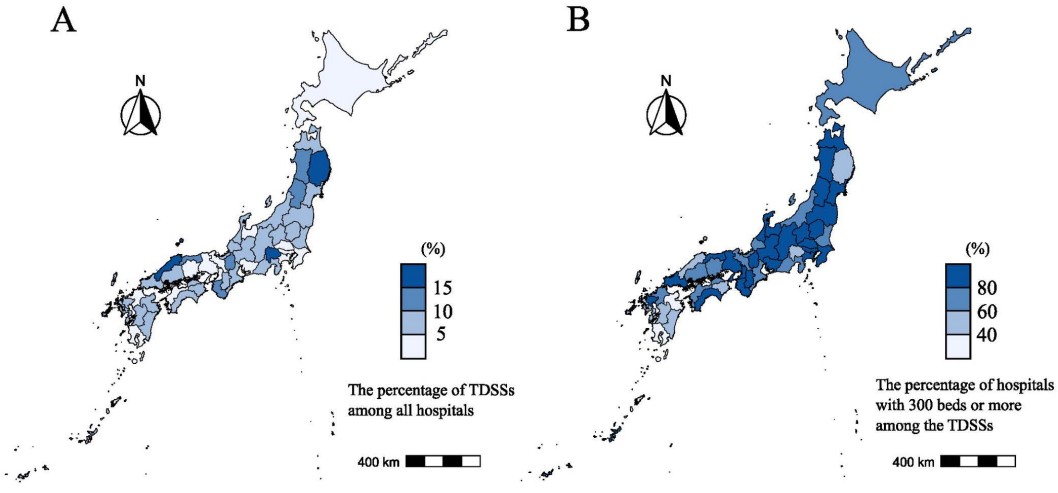

**Fig 1. Distribution of 463 TDSSs.** A. The percentage of TDSSs among all hospitals in each prefecture. B. The percentage of hospitals with 300 beds or more among the TDSSs in each prefecture. The blue color scale indicates the distribution across each percentage range. We created the map by editing data 'Global Map Japan'. TDSS, target designated sentinel site. Source: 'Global Map Japan (Technical Report of the Geospatial Information Authority of Japan, D1-No.576)'(Geospatial Information Authority of Japan) (https://www.gsi.go.jp/kankyochiri/gm_japan_e.html) PDL1.0(https://www.gsi.go.jp/kika-kuchousei/kikakuchousei40182.html) (accessed on July 7, 2025).

**Table 2. Distribution of TDSSs that responded to the questionnaire.**

| Types of TDSSs | Number of TDSSs | TDSSs responding to the questionnaire n, (%) | | TDSSs responding appropriately to the questionnaire n, (%) | |
|---|---|---|---|---|---|
| All TDSSs | 463 | 234 | (50.5) | 231 | (49.9) |
| TDSSs with 300 or more beds | 347* | 191 | (55.0) | 188 | (54.2) |
| TDSSs with fewer than 300 beds | 116† | 43 | (37.1) | 43 | (37.1) |

TDSS, target designated sentinel site.

*Accounts for 24.2% of all institutions in Japan with 300 or more beds [32].

†Accounts for 1.7% of all institutions in Japan with fewer than 300 beds [32].

and 69 underreported as MDRP infections (S5 Table), with Fig 3 showing the distribution of TDSSs that misreported and underreported MDRP infections. Two TDSSs with fewer than 300 beds misreported 10 and 11 cases, respectively, and one TDSS with 300 or more beds underreported seven cases of accurate MDRP infections. We excluded these cases as outliers when calculating PPV and sensitivity. Thus, the PPV and sensitivity for the reports of MDRP infections to NESID between 2018 and 2022 were 0.45 and 0.65, respectively (Table 3).

## Qualitative evaluation

One public health officer working at a local infectious disease surveillance center said that DSSs were selected properly based on Japanese law. However, this same public health officer, some of the physicians, and one infection control nurse pointed out that it was insufficient for only DSSs to report MDRP infections because outbreaks occurring at long-term care facilities or small hospitals may be overlooked. Additionally, they questioned the appropriateness of reporting MDRP infections monthly because such outbreaks require prompt public health action. Some physicians found it easy to report the cases to NESID, but these physicians and clinical microbiology technologists mentioned that reporting errors may be

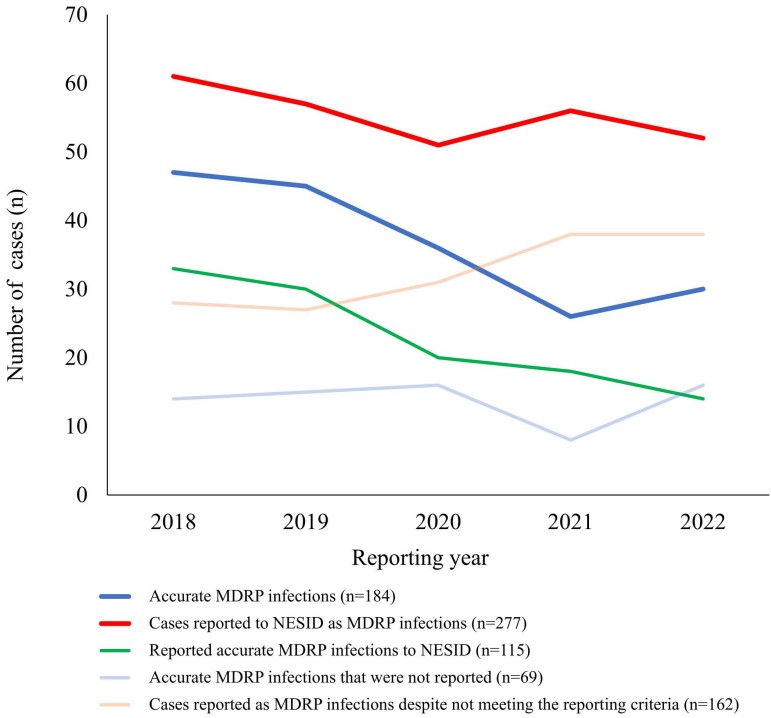

Legend:
- Accurate MDRP infections (n=184)
- Cases reported to NESID as MDRP infections (n=277)
- Reported accurate MDRP infections to NESID (n=115)
- Accurate MDRP infections that were not reported (n=69)
- Cases reported as MDRP infections despite not meeting the reporting criteria (n=162)

**Fig 2. Trend in the number of cases reported as MDRP infections and the number of accurate MDRP infections at TDSSs, 2018–2022.** From 2018 to 2022, these TDSSs reported 277 cases as MDRP infections, while 184 cases were accurately reported MDRP infections, with both numbers decreasing over time. NESID, National Epidemiological Surveillance of Infectious Diseases; MDRP, multidrug-resistant *Pseudomonas aeruginosa*; TDSS, target designated sentinel site.

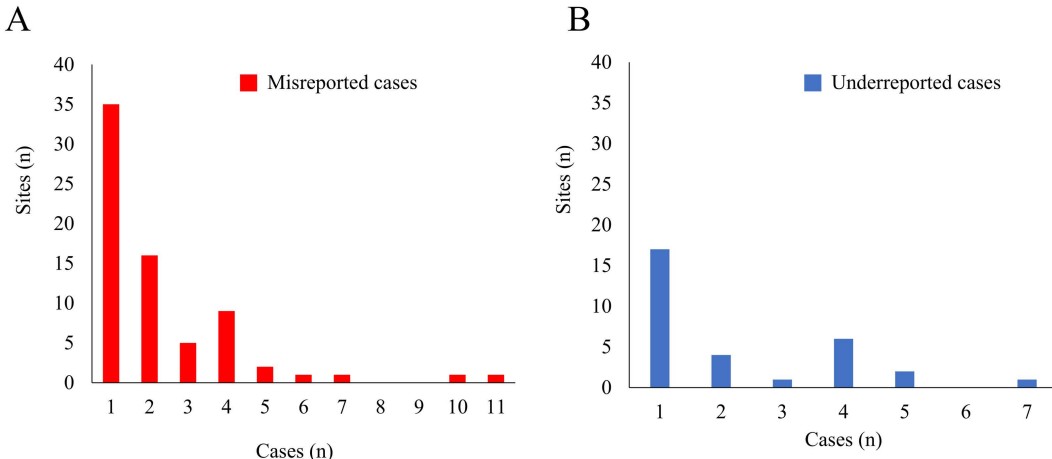

**Fig 3. Distribution of TDSSs that misreported cases as MDRP infections and underreported accurate MDRP infections, 2018-2022.** A. Seventy-one TDSSs misreported a total of 162 cases as MDRP infections despite them not meeting the criteria. B. Thirty-one TDSSs underreported a total of 69 accurate MDRP infections. MDRP, multidrug-resistant *Pseudomonas aeruginosa*; TDSS, target designated sentinel site.

**Table 3. Positive predictive value and sensitivity for reports of MDRP infections between 2018 and 2022.**

| Notifications by types of TDSSs | Positive predictive value | Sensitivity |
|---|---|---|
| Notification from all TDSSs | 0.45 | 0.65 |
| Notification from TDSSs with 300 or more beds | 0.48 | 0.64 |
| Notification from TDSSs with fewer than 300 beds | 0.19 | 1.00 |

MDRP, multidrug-resistant *Pseudomonas aeruginosa*; TDSS, target designated sentinel site.

caused by complicated criteria used to confirm resistance to three antibiotics using standards different from the break-points of CLSI and EUCAST, and colonized cases were excluded. These physicians and national academic authorities indicated that the reports of CRPA infections would enable comparison of the situation with other countries and could be utilized as an indicator for the appropriate usage of broad-spectrum antibiotics based on the Japanese NAP. For these reasons, most of the interviewees concluded that the current surveillance system for MDRP infections is difficult for the government and medical institutions to utilize for the control of drug-resistant *P. aeruginosa*.

### The advantages and disadvantages of possible surveillance systems of drug-resistant *P. aeruginosa* infections

We have summarized the advantages and disadvantages of sentinel surveillance and notifiable surveillance for both MDRP and CRPA as possible surveillance systems (Table 4).

## Discussion

We evaluated the Japanese surveillance system, NESID, for surveillance of MDRP infections quantitatively and qualitatively based mainly on the CDC guidelines [29]. Although current surveillance partially captured the trend of MDRP infections in Japan, the surveillance should be designed based on actions and goals for the control of drug-resistant *P. aeruginosa* infections in the country.

The number of accurate MDRP infections occurring at TDSSs decreased over time, as did the number of cases reported as MDRP infections. Moreover, NESID showed good representativeness because TDSSs uniformly existed in each prefecture according to the population [20], and there was no geographical bias detected in our findings. Thus, we considered that the decreasing trend in surveillance data in TDSSs reflected the actual trend of MDRP infections across Japan. Additionally, data from JANIS, another national-level AMR surveillance system in Japan, also showed a decreasing trend of MDRP detection in inpatient settings from 0.05% in 2017 to 0.02% in 2023, which may also reflect the actual domestic situation due to its high coverage of hospitals (close to one-third of all hospitals participated in 2023) in Japan [19,23]. We were unable to distinguish between nosocomial infections (typically defined as infections occurring at least 48 hours after admission in Japan) and community-acquired infections based on the NESID data. However, since the JANIS data showed that MDRP was detected more frequently in hospitals than in outpatient settings [23,24], it is likely that most MDRP infections reported to NESID occurred in hospitals. Consequently, the original objective of NESID to assess the trend of MDRP infections has been achieved, and we confirmed from the surveillance data that the number of MDRP infections in Japan has been decreasing.

We learned through the interviews that some of the interviewees were of the opinion that the current surveillance system does not meet public health needs. Thus, the current situation should be reviewed to improve the surveillance system to meet new objectives in the control of drug-resistant *P. aeruginosa* infections nationwide. Drug-resistant *P. aeruginosa* was classified into CRPA, MDRP, extensively drug-resistant *P. aeruginosa* (XDRP), and pan drug-resistant *P. aeruginosa* (PDRP) according to the standardized international terminology published in 2012 [33]. Furthermore, difficult-to-treat resistance *P. aeruginosa* (DTRP) was defined in 2018, with a focus on the lack of effective first-line drugs [34–36]. As drug-resistant *P. aeruginosa* has been classified in several ways, each country focuses on different types [37]. The important

**Table 4. Characteristics of candidate surveillance systems for drug-resistant *Pseudomonas aeruginosa*.**

| Types of drug-resistant *P. aeruginosa* | | Purpose | Advantages | Disadvantages |
|---|---|---|---|---|
| MDRP | Sentinel surveillance | • To evaluate the trend in the number of reported MDRP infections from sentinel sites | • Higher concern from the public than for CRPA | • Requires confirmation of resistance to three anti-pseudomonas antibiotics, which may lead to reporting errors<br>• Difficulty in detecting outbreaks at non-sentinel sites<br>• Difficulty in comparing the situation of AMR with other countries due to different definitions of MDRP |
| | Notifiable disease surveillance | • To detect outbreaks to strengthen infection control measures<br>• To evaluate the trend in the number of notified MDRP infections from all facilities | • Higher concern from the public than for CRPA<br>• Ability to detect outbreaks at all facilities including non-sentinel sites | • Requires confirmation of resistance to three anti-pseudomonas antibiotics, which may lead to reporting errors<br>• Difficulty in comparing the situation of AMR with other countries due to different definitions of MDRP |
| CRPA | Sentinel surveillance | • To evaluate the trend in the number of reported CRPA infections from sentinel sites | • Confirmation of resistance to only carbapenems allows for a relatively straightforward determination<br>• Ability to easily compare the situation of AMR with other countries<br>• Ability to focus on and evaluate carbapenemase | • Difficulty in detecting outbreaks at non-sentinel sites<br>• Burden on medical institutions due to increase in the number of reports more than for MDRP |
| | Notifiable disease surveillance | • To detect outbreaks to strengthen infection control measures<br>• To evaluate the trend in the number of notified CRPA infections from all facilities | • Confirmation of resistance to only carbapenems allows for a relatively straightforward determination<br>• Ability to detect outbreaks in all facilities including non-sentinel sites<br>• Ability to easily compare the situation of AMR with other countries<br>• Ability to focus on and evaluate carbapenemase | • Severe burden on medical institutions due to excessive increase in number of notification |

AMR, antimicrobial resistance; CRPA, carbapenem-resistant *P. aeruginosa*; MDRP, multidrug-resistant *P. aeruginosa*.

points to be discussed in designing a surveillance system for drug-resistant *P. aeruginosa* are the definition of drug resistance including antimicrobial susceptibility testing and whether to conduct notifiable diseases surveillance or sentinel surveillance.

MDRP is defined as *P. aeruginosa* non-susceptible to at least one agent in three and over antimicrobial categories such as carbapenems [33]. MDRP has quite limited treatment options and often leads to high inpatient mortality and prolonged hospital stays [11,38]. Thus, early detection is essential for an effective response to MDRP outbreaks. However, not all MDRP outbreaks can be identified through sentinel surveillance unless the outbreak is epidemiologically or genetically unusual [39]. To respond to MDRP outbreaks promptly for its elimination, notifiable disease surveillance is preferable to sentinel surveillance because notifiable disease surveillance can detect MDRP infections from all hospitals. In addition, although case fatality is difficult to calculate due to insufficient outcome information, notifiable disease surveillance enables us to calculate the incidence and proportion of nosocomial infections if almost all the cases were reported and the information was comprehensive. One critical challenge of the notifiable disease surveillance of MDRP is the difficulty in confirming drug resistance. Our results revealed an increase in the number of cases reported as MDRP infections despite not meeting the reporting criteria from 2018 to 2022. This issue would persist even if the surveillance system were changed to notifiable disease surveillance. Thus, quality assurance of antimicrobial susceptibility testing is the key to the

successful implementation of the notification system. Another challenge is that we need to note whether the definition of MDRP is consistent when we compare the proportions of MDRP with other countries and regions.

Many developed countries monitor CRPA infections, as carbapenems are essential broad-spectrum antibiotics. Such monitoring enables these countries to compare the carbapenem resistance rates of *P. aeruginosa* with each other [40]. For instance, EARS-Net determined the proportions of CRPA in each country within the WHO European Region and showed large differences [16]. A previous article indicated that appropriate use of carbapenems was inversely correlated with the prevalence of CRPA [41]; thus, the proportions of CRPA could serve as an indicator of the rational use of carbapenems. In addition, as carbapenemase-producing isolates of *P. aeruginosa* can cause serious outbreaks [7,42], laboratory testing for carbapenemase is crucial. In the U. S., the AR Lab Network reported detecting carbapenemase genes in 2.5% of 86,878 CRPA isolates analyzed between 2017 and 2023, with 61.0% of them identified as VIM [18]. In contrast, only 4.2% of 382 meropenem-resistant CRPA isolates collected from 78 hospitals in Japan possessed acquired carbapenemases, with the IMP type being the most frequent, and no VIM-producing CRPA isolates were detected. Multilocus sequence typing of the 382 CRPA isolates revealed that ST274 was predominant. Notably, ST274 CRPA isolates rarely carried carbapenemase [43]. We consider it important to integrate the reporting system with laboratory-based surveillance to comprehensively evaluate the trend of CRPA infections, including the results of genomic sequencing. However, if CRPA infections are classified as notifiable diseases, excessive numbers of cases would be reported, potentially leading to economic costs and burdens for medical and laboratory institutions because *P. aeruginosa* has resistance to carbapenems through various mechanisms other than carbapenemase production [8,9]. Thus, sentinel surveillance integrated with laboratory-based surveillance would be indispensable to controlling carbapenem resistance by assessment of the trend of the number of CRPA infections and evaluation of whether CRPA isolates produce any of the carbapenemases.

The strength of our study is that it is, to our knowledge, the first report to comprehensively evaluate surveillance systems for drug-resistant *P. aeruginosa* infections and to discuss the advantages and disadvantages of potential candidate systems. However, our study also has some limitations. First, as we interviewed staff members working at hospitals and one local infectious disease surveillance center located in a specific prefecture, it is possible that the situation might differ somewhat from those in other regions. However, we endeavored to generalize the findings by conducting interviews with national academic authorities from various prefectures. Moreover, although the total number of participants was limited to 17, the interviewees included infectious diseases specialists and individuals from a variety of professional backgrounds. Second, the interview results could be influenced both by social desirability bias that interviewees' facilities have fewer underreported or misreported cases and by recall bias. Thus, we performed not only qualitative but also quantitative analysis for a comprehensive evaluation. Third, although there have been no changes in the national surveillance system for MDRP infection itself, changes in the commitment and efficiency of those in charge of surveillance at individual medical institutions may have occurred over the years. Fourth, the reports of MDRP infections in NESID and our questionnaire survey were not confirmed by laboratory testing.

## Conclusion

Our evaluation of the surveillance system for MDRP infections in Japan indicated that the reports of MDRP infections from DSSs represented the nationwide trend in MDRP infections, which has been decreasing over time. However, our study also showed that this system did not seem to meet current public health needs. Of concern were the difficulties in timely detection of and response to a cluster of MDRP infections due to reporting at monthly intervals, inaccurate reporting due to complex diagnostic criteria, and in international comparison due to different susceptibility definitions. As the epidemiology of drug-resistant *P. aeruginosa* is changing, the national policy may differ according to the public health needs, and the surveillance system also needs to be modified. Potential candidate systems may be a notifiable disease surveillance for MDRP infections or a sentinel surveillance integrated with laboratory-based surveillance for CRPA infections. On the basis of our findings, we can contribute to the development of the refined surveillance system to better manage drug resistance in *P. aeruginosa*.

## Supporting information

**S1 Fig. Information flow of surveillance system for MDRP in Japan.** The sentinel sites report the MDRP infections with the information of age, sex, and detected specimen to the local public health center monthly. The central infectious disease surveillance center integrates the data and shares them with the Ministry of Health, Labour and Welfare. MDRP, multidrug-resistant *Pseudomonas aeruginosa*.
(TIF)

**S2 Fig. Theoretical scheme of cases misreported as MDRP infections and underreported as accurate infections.** (i) The number of cases reported as MDRP infections. (ii) The accurate number of MDRP infections. (iii) The number of cases misreported as MDRP infections. (iv) The number of underreported accurate MDRP infections. MDRP, multidrug-resistant *Pseudomonas aeruginosa*.
(TIF)

**S3 Fig. Trend in the number of cases reported to NESID as MDRP infections by age group between 2013 and 2022.** The number of reported cases of MDRP infection showed a decreasing trend in both males (A) and females (B), especially among the elderly. MDRP, multidrug-resistant *Pseudomonas aeruginosa*; NESID, National Epidemiological Surveillance of Infectious Diseases.
(TIF)

**S1 Table. Criteria related to drug resistance in *Pseudomonas aeruginosa*.**
(XLSX)

**S2 Table. Distribution of age group and sex in reports of MDRP infections to NESID from 347 DSSs.**
(XLSX)

**S3 Table. Dataset on the distribution of TDSSs in Japan.**
(XLSX)

**S4 Table. Dataset associated with the response to the questionnaire survey from 234 TDSSs.**
(XLSX)

**S5 Table. Comparison data between the number of MDRP infections reported to NESID by 231 TDSSs from 2018 to 2022 and the number of MDRP infections reported as accurate MDRP infections on the questionnaire survey.**
(XLSX)

**S1 Appendix. Questionnaire on the testing and reporting system for multidrug-resistant *Pseudomonas aeruginosa* (MDRP).**
(DOCX)

**S2 Appendix. Interview guides of surveillance evaluation for multi-drug resistant *Pseudomonas aeruginosa* (MDRP).**
(DOCX)

## Acknowledgments

We are grateful to public health officers, physicians, nurses, pharmacists, and clinical microbiology technologists who participated in the face-to-face interviews. We also thank the staff working at the designated sentinel sites, local public health centers, local infectious disease surveillance centers, and local public health institutes for cooperating with reporting to the national surveillance system and for responding to our questionnaire survey.

## Author contributions

**Conceptualization:** Shogo Otake, Takuya Yamagishi.

**Data curation:** Shogo Otake, Takuya Yamagishi.

**Formal analysis:** Shogo Otake, Takuya Yamagishi.

**Funding acquisition:** Takuya Yamagishi.

**Investigation:** Shogo Otake, Takuya Yamagishi.

**Methodology:** Takuya Yamagishi.

**Project administration:** Takuya Yamagishi.

**Resources:** Takuya Yamagishi.

**Supervision:** Takuya Yamagishi, Kandai Nozu, Tomimasa Sunagawa.

**Writing – original draft:** Shogo Otake, Takuya Yamagishi.

**Writing – review & editing:** Takuya Yamagishi, Takayuki Shiomoto, Manami Nakashita, Hitomi Kurosu, Chiaki Ikenoue, Hirofumi Kato, Munehisa Fukusumi, Tomoe Shimada, Takuri Takahashi, Motoi Suzuki, Teruo Kirikae, Yoshichika Arakawa, Kandai Nozu, Tomimasa Sunagawa, Motoyuki Sugai.

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
