## [Decision Letter · Decision Letter 0]

26 Mar 2025

PONE-D-25-07682What we can and cannot see from the surveillance for drug-resistant Pseudomonas aeruginosa - Findings from the evaluation of surveillance system for multidrug-resistant P. aeruginosa infections in Japan -PLOS ONE

Dear Dr. Yamagishi,

Thank you for submitting your manuscript to PLOS ONE. After careful consideration, we feel that it has merit but does not fully meet PLOS ONE’s publication criteria as it currently stands. Therefore, we invite you to submit a revised version of the manuscript that addresses the points raised during the review process.

We look forward to receiving your revised manuscript.

Kind regards,

Mabel Kamweli Aworh, DVM, MPH, PhD. FCVSN

Academic Editor

PLOS ONE

Journal Requirements:

Grant from Ministry of Health, Labour, and Welfare, Japan

5. We note that you have indicated that there are restrictions to data sharing for this study. For studies involving human research participant data or other sensitive data, we encourage authors to share de-identified or anonymized data. However, when data cannot be publicly shared for ethical reasons, we allow authors to make their data sets available upon request. For information on unacceptable data access restrictions, please see http://journals.plos.org/plosone/s/data-availability#loc-unacceptable-data-access-restrictions.

6. In the online submission form, you indicated that data we collected can be shared publicly, but data on the National Epidemiological Surveillance of Infectious Diseases cannot be shared publicly because the data are strictly protected by the law and need for the governmental clearance for use.

7. We note that Figure 1 in your submission contain [map/satellite] images which may be copyrighted. All PLOS content is published under the Creative Commons Attribution License (CC BY 4.0), which means that the manuscript, images, and Supporting Information files will be freely available online, and any third party is permitted to access, download, copy, distribute, and use these materials in any way, even commercially, with proper attribution. For these reasons, we cannot publish previously copyrighted maps or satellite images created using proprietary data, such as Google software (Google Maps, Street View, and Earth). For more information, see our copyright guidelines: http://journals.plos.org/plosone/s/licenses-and-copyright.

Additional Editor Comments :

In addition to addressing the reviewer’s comments, please ensure the following issue is resolved:

The reference list must be updated to include more recent studies published within the last five years. Currently, older references constitute 43% of the list, which exceeds the acceptable limit of 20%. This needs to be rectified to meet the required standards.

Reviewers' comments:

Reviewer's Responses to Questions

**Comments to the Author**

1. Is the manuscript technically sound, and do the data support the conclusions?

Reviewer #1: Yes

Reviewer #2: Yes

Reviewer #3: Yes

2. Has the statistical analysis been performed appropriately and rigorously? 

Reviewer #1: Yes

Reviewer #2: Yes

Reviewer #3: Yes

3. Have the authors made all data underlying the findings in their manuscript fully available?

Reviewer #1: Yes

Reviewer #2: No

Reviewer #3: Yes

4. Is the manuscript presented in an intelligible fashion and written in standard English?

Reviewer #1: Yes

Reviewer #2: Yes

Reviewer #3: Yes

5. Review Comments to the Author

Reviewer #1: The manuscript seeks to review the current surveillance reporting system for multi drug resistant Pseudomonas aeruginosa. The authors utilize both quantitative measures (reported case counts) and qualitative measures using questionnaires and interviews. The authors provide clear exclusion criteria for further analysis and provide a clear analysis of the results.

The interviewee size for the interview section is not mentioned, we do not know if this size is small, this limits the generalizability of the section. it is great that the authors clearly discuss the other limitations with their interviews, such as the fact that most interviewees were from one prefecture, I think the sample size is a key datapoint that should be mentioned. For example, if you interview only 50 individuals and the actual no of potentially appropriate interviewees is 5000, the sample size is too small to make any claims about what the true needs are.

Also, some of the interview questions are binary, showing the actual count of respondents that favored one way or the other would provide quantitative data for government future efforts to improve. For example: "IX. Which do you think is better as a reportable disease: MDRP infections or carbapenem-resistant P. aeruginosa infections?" If 70% of respondents prefer CARP to MDRP, then it is clear the majority have this preferences vs say 51%. What is currently reported does not provide such pertinent details.

Finally, While the manuscript is fairly well written, the main text figures and table legends appear slapped in between the results and discussions section. There are also a few small grammar errors that need to be addressed with the writing.

Providing these additional details will strengthen the findings of this paper and provide useful information for improving the system.

Reviewer #2: I must commend the authors for conducting this important evaluation on the surveillance of multidrug-resistant P. aeruginosa infections in Japan. The topic is highly relevant to public health and antimicrobial resistance (AMR) control efforts. The manuscript is well-structured, and the findings provide valuable insights into the strengths and weaknesses of the current surveillance system. While the study is strong overall, there are a few areas that could be clarified or expanded to further enhance the manuscript.

ABSTRACT

Line 38: Replace the word "the" to "this" to enhance the clarity

Line 40: It is important to highlight the key indicators assessed here in the methods.

Line 40: it is also unclear from this abstract what the geographical scope is. can you highlight this too?

Line 41: Its not clear what guidelines are been referred to here. Please be more specific. Is there a universal guideline for conducting cross-sectional studies?

Line 42: It is essential to start by mentioning that this was a "mixed method study" before stating the quantitaive and qualitative components. This will aid better understanding of the methods

Line 45: why was the quantitative aspect of this study done for different dates? 2013 and 2018? why not both arms of the study conducted from 2013 to 2022?

Line 45: can you specify the kind of interviews that were conducted. Were they Key Informant interviews or Indepth interviews or....?

Line 61: The word "may" should be replaced with "would" as this is a recommendation.

INTRODUCTION

Lines 61 - 117: This introduction is great. However, it is important to include some information concerning the context of AMR in Japan generally and .

- can you provide the burden of AMR in Japan, particularly in healthcare facilities and community settings?

- It may be also important to highlight in this introduction, any challenges to the existing surveillance efforts (e.g., underreporting, limited laboratory capacity, lack of standardized protocols e.t.c). this will help to paint a better picture of the problem. Because as it stands, your focus is on whether the downward trend accurately represents the actual trend in MDRP infections but little is said about existing challenges which could be linked to this trend. Are there previous studies or reports on MDR P. aeruginosa infections in Japan?

Lines 117 - 119: Are there existing documented gaps in the current surveillance system for MDR P. aeruginosa to justify this evaluation?(e.g., lack of real-time data, poor integration with hospital reporting systems e.t.c). This is important to justify why evaluating the surveillance system is essential for strengthening AMR monitoring and improving infection control strategies not just only because you want to know whether the trend accurately represents actual trend.

Lines 122 - 124: Clearly state the main objective of this study here.

is the objective to “evaluate the effectiveness, validity, sensitivity, and limitations of the surveillance system for multidrug-resistant Pseudomonas aeruginosa infections in Japan"?

Optionally, break it down into specific objectives, such as:

- Assessing the completeness, timeliness, and accuracy of surveillance data.

- Identifying gaps in case detection and reporting.

e.t.c

the way it is written, its not clear what the objectives are.

METHODS

Line 126: SHould be "methods" not "method"

Line 129: more accurately put as "Mixed methods"

Line 129 - 134: The Aim should not be in the methods but in the introduction or as a separate subsection before the methods. Please revise accordingly

Lines 144 - 157: It is important to provide more specific context about the Study Setting and Population to allow better understanding of the results

- please describe the geographical scope (e.g., national, selected states, or specific healthcare facilities in Japan).

- Mention the type of healthcare facilities involved (e.g., tertiary hospitals, secondary health centers, surveillance units).

- Define the target population (e.g., patients with laboratory-confirmed MDR P. aeruginosa infections, healthcare providers, surveillance officers).

Line 163: can you describe where this data is stored? from where was it accessed?

LIne 163: it is important to provide more details of this questionnaire

- was it adopted or adapted (or even developed by the researchers) - please include reference for the source

- how was the questionnaire validated?

- who were the respondents of the questionnaire

- what language was the questionnaire in?

- what form was the questionnaire (paper or electronic) - if electronic, was a certain platform or tool used e.g - Goodle forms, microsoft forms, Kobotoolbox e.t.c)

- who were the respondents??????

Line 180; A questionnaire is not typically used for interviews. Qualitative data is not usually collected using questionnaires. can you explain what type of interview was done and why you decided to use a questionnaire instead of an interview guide?

Line 182: - where were these physicians and technologists based?

- can you provide more details about the tool used for the interview and how the qualitative data was collected. was this audio recorded? what tools were used? Was the interview structured or unstructured. It is important to be thoruogh with your methodology to allow reproducibility and clarity

Line 184 - 197: There should be a separate subsection (not Analysis) to define the variables collected, such as the surveillance indicators (e.g., completeness, timeliness, sensitivity, specificity). This should not come under analysis

Under analysis section you should describe the statistical methods used to analyze quantitative data (e.g., descriptive statistics, proportions, trend analysis). It is also important to mention any qualitative analysis conducted (e.g., thematic analysis for key informant interviews). This section is completely silent on how you analysed the qualitative data

please specify the software used for data analysis (e.g., SPSS, Stata, R, NVivo e.t.c).

RESULTS

Line 238: Table 2- please indicate the appropriate column header here.

Line 256: Table 3- please indicate the appropriate column header here.

Line 259: Qualitative findings are typically presented under thematic headings, supported by direct quotes from respondents. There is no description in the methods as to how the qualitative data was collected and analysed. Clearly state how qualitative data were analyzed (e.g., thematic analysis using NVivo or manual coding). Also, describe the process of identifying themes, including how transcripts were reviewed, coded, and categorized. please reconcile with the methods.

DISCUSSION

Line 289: Can you compare your findings to similar studies from Japan, Asia, or global surveillance reports

Line 298: its not clear how you triangulated the results from the quantitative with your qualitative results. can you explain the areas of overlap?

Line 311 - 314: where did this data come from? did you collect this information from existing literature or you did a review? Please include the reference for the sources of these information. What part of your methods covered this area? please clarify

Line 368: what are the strengths of your study? This would be a good place to include them.

Map: This is a good map. However, it is important to include the scale as well as cardinal points/North arrow. Clearly label the legend too.

Reviewer #3: The paper is well written and should be published. The use of longitudinal quantitative data over a wide period of time is commendable. And combining this with questionnaire and interview data makes it robust.

Line 83. The U.S. Centers for Disease Control and Prevention (CDC) has estimated 32,600 cases and 2,700 deaths due to

MDRP infections annually and classified them as a serious public health threat, calling for the need for prompt and sustained action in 2019 [11]. Can the authors provide more Japan specific data vis a vis MDRP epidemiology?

For the results and observations prior to this study, if possible, can the authors provide a demographic breakdown of how MDRP affects the population- age, sex, immune status, etc. How many of these MDRPs were community or nosocomially acquired?

Can the authors provide data or explain how the decreasing trend they observed is true and not a product of some unaccounted for confounders? Is it not possible that give the period of time covered by the study, that there might have been a change in the commitment and efficiency of those in charge of the surveillance over the years? Depending on the answers, these new insights could be added to the limitations.

6. PLOS authors have the option to publish the peer review history of their article (what does this mean?). If published, this will include your full peer review and any attached files.

Reviewer #1: No

Reviewer #2: **Yes: **Abdulhakeem Abayomi Olorukooba

Reviewer #3: No

---

## [Author Response · Author response to Decision Letter 1]

10 May 2025

We attached the response as a file because it was long.

---

## [Decision Letter · Decision Letter 1]

27 May 2025

PONE-D-25-07682R1What we can and cannot see from the surveillance for drug-resistant Pseudomonas aeruginosa - Findings from the evaluation of surveillance system for multidrug-resistant P. aeruginosa infections in Japan -PLOS ONE

Dear Dr. Yamagishi,

Thank you for submitting your manuscript to PLOS ONE. After careful consideration, we feel that it has merit but does not fully meet PLOS ONE’s publication criteria as it currently stands. Therefore, we invite you to submit a revised version of the manuscript that addresses the points raised during the review process.

We look forward to receiving your revised manuscript.

Kind regards,

Mabel Kamweli Aworh, DVM, MPH, PhD. FCVSN

Academic Editor

PLOS ONE

Journal Requirements:

Please review your reference list to ensure that it is complete and correct. If you have cited papers that have been retracted, please include the rationale for doing so in the manuscript text, or remove these references and replace them with relevant current references. Any vidichanges to the reference list should be mentioned in the rebuttal letter that accompanies your revised manuscript. If you need to cite a retracted article, indicate the article’s retracted status in the References list and also include a citation and full reference for the retraction notice.

Additional Editor Comments:

1. Line 59 - Please change "Discussion" to "Conclusion" in the abstract.

**2. Discussion Section (Line 322 and beyond):**

Please remove the citation to “Fig 1” from line 322, "S2 Table" from line 344, and "Table 4" from line 348 in the Discussion section. In accordance with journal guidelines, tables and figures should not be cited in the Discussion section, which should focus on interpreting the results and offering plausible explanations rather than repeating findings. Additionally, please relocate **Table 4** and the accompanying texts to the Results section, where it is more appropriately discussed. Please make sure to cite Table 4 in the Results section. Lastly, for improved flow and clarity, we recommend removing subheadings within the Discussion section.

Reviewers' comments:

Reviewer's Responses to Questions

**Comments to the Author**

1. If the authors have adequately addressed your comments raised in a previous round of review and you feel that this manuscript is now acceptable for publication, you may indicate that here to bypass the “Comments to the Author” section, enter your conflict of interest statement in the “Confidential to Editor” section, and submit your "Accept" recommendation.

Reviewer #2: All comments have been addressed

Reviewer #3: (No Response)

2. Is the manuscript technically sound, and do the data support the conclusions?

Reviewer #2: Yes

Reviewer #3: Yes

3. Has the statistical analysis been performed appropriately and rigorously? 

Reviewer #2: Yes

Reviewer #3: Yes

4. Have the authors made all data underlying the findings in their manuscript fully available?

Reviewer #2: Yes

Reviewer #3: (No Response)

5. Is the manuscript presented in an intelligible fashion and written in standard English?

Reviewer #2: Yes

Reviewer #3: Yes

6. Review Comments to the Author

Reviewer #2: The authors have done a great job and have addressed all my concerns and i dont have any other comments on this manuscript.

Reviewer #3: You have stated that "the proportion of MDRP detected per number of patients who submitted specimens in 2023 was 0.004% (134/3,441,898 samples) for outpatient specimens (Reference 24), and 0.02% (651/3,089,112 samples) for inpatient specimens (Reference 23), which is approximately five times higher. Therefore, we inferred that hospital-acquired infections likely account for a greater proportion of MDRP infections as well." To further clarify for readers, do state the working definitions and cut off for differentiating between hospital and community acquired infections in Japan and expand the discussion as to how health care facilities seem to be a breeding house for MDRP according to your data. Kindly proffer recommendations

Also, recommend what can be done to improve collection of data related to impacts of MDRP infections including MDRP infection related deaths in Japan.

7. PLOS authors have the option to publish the peer review history of their article (what does this mean?). If published, this will include your full peer review and any attached files.

Reviewer #2: No

Reviewer #3: No

---

## [Author Response · Author response to Decision Letter 2]

7 Jul 2025

Response to the editorial comments

1. Line 59 - Please change "Discussion" to "Conclusion" in the abstract.

Response: We have changed the word accordingly. (Line 59)

2. Discussion Section (Line 322 and beyond):

Please remove the citation to “Fig 1” from line 322, "S2 Table" from line 344, and "Table 4" from line 348 in the Discussion section. In accordance with journal guidelines, tables and figures should not be cited in the Discussion section, which should focus on interpreting the results and offering plausible explanations rather than repeating findings. Additionally, please relocate Table 4 and the accompanying texts to the Results section, where it is more appropriately discussed. Please make sure to cite Table 4 in the Results section. Lastly, for improved flow and clarity, we recommend removing subheadings within the Discussion section.

Response: Thank you for your recommendations. We have removed the citation to “Fig 1”, "S2 Table", and "Table 4" in the Discussion section, and have also removed subheadings within that section. In addition, following your suggestions, we have relocated Table 4 and the accompanying texts to the Results section as shown below.

“The advantages and disadvantages of possible surveillance systems of drug-resistant P. aeruginosa infections

We have summarized the advantages and disadvantages of sentinel surveillance and notifiable surveillance for both MDRP and CRPA as possible surveillance systems (Table 4).” (Lines　312-316)

Additionally, due to the frequent updates of national action plans, we decided to exclude S2 Table, and the column named “Countries including target benchmark for resistance of P. aeruginosa in each NAP [36]” in Table 4 to guarantee the accuracy of the information.

Reviewer's comments:

Reviewer #3:

1. You have stated that "the proportion of MDRP detected per number of patients who submitted specimens in 2023 was 0.004% (134/3,441,898 samples) for outpatient specimens (Reference 24), and 0.02% (651/3,089,112 samples) for inpatient specimens (Reference 23), which is approximately five times higher. Therefore, we inferred that hospital-acquired infections likely account for a greater proportion of MDRP infections as well." To further clarify for readers, do state the working definitions and cut off for differentiating between hospital and community acquired infections in Japan and expand the discussion as to how health care facilities seem to be a breeding house for MDRP according to your data. Kindly proffer recommendations

Response: Thank you for your valuable suggestion. In Japan, for surveillance of healthcare associated infections, medical institutions generally define infections that occur more than 48 hours after admission as nosocomial (or healthcare-associated) infections. However, unlike notifiable diseases such as carbapenem-resistant Enterobacteriaceae infections, sentinel surveillance does not require information on whether infections are nosocomial or community-acquired, nor does it collect data on whether specimens were obtained in outpatient or inpatient settings.

From different perspective, the meropenem resistance of P. aeruginosa was 8.8% in inpatient specimens compared to 2.6% in outpatient specimens in 2023 in JANIS data (which discloses the classification of specimen collection locations), suggesting that CRPA infections are also more likely to occur in hospital settings. Therefore, although it is difficult to distinguish between nosocomial and community-acquired infections from the NESID data, it is likely that the most MDRP infections reported to NESID were occurring in hospitals.

Thus, we have added the following sentences in the Discussion section.

“We were unable to distinguish between nosocomial infections (typically defined as infections occurring at least 48 hours after admission in Japan) and community-acquired infections based on the NESID data. However, since the JANIS data showed that MDRP was detected more frequently in hospitals than in outpatient settings [23,24], it is likely that most MDRP infections reported to NESID occurred in hospitals.” (Lines 334-339)

2. Also, recommend what can be done to improve collection of data related to impacts of MDRP infections including MDRP infection related deaths in Japan.

Response: Thank you for your valuable recommendation. If MDRP infections were reported through notifiable disease surveillance, the incidence and proportion of nosocomial infections could be calculated. However, information on MDRP infection-related deaths is difficult to evaluate in national surveillance, where many cases are not followed up. We believe that a well-designed cohort study is necessary to evaluate the fatality. Thus, we have revised and added the sentences as follows.

“To respond to MDRP outbreaks promptly for its elimination, notifiable disease surveillance is preferable to sentinel surveillance because notifiable disease surveillance can detect MDRP infections from all hospitals. In addition, although case fatality risk is difficult to calculate due to insufficient outcome information, notifiable disease surveillance enables us to calculate the incidence and proportion of nosocomial infections if cases were reported with a sufficient capture rate and the information was comprehensive.” (Lines 361-367)

---

## [Editor Report · Decision Letter 2]

21 Jul 2025

What we can and cannot see from the surveillance for drug-resistant Pseudomonas aeruginosa - Findings from the evaluation of surveillance system for multidrug-resistant P. aeruginosa infections in Japan -

PONE-D-25-07682R2

Dear Dr. Yamagishi,

We’re pleased to inform you that your manuscript has been judged scientifically suitable for publication and will be formally accepted for publication once it meets all outstanding technical requirements.

Kind regards,

Mabel Kamweli Aworh, DVM, MPH, PhD. FCVSN

Academic Editor

PLOS ONE